# Associations between active travel and diet: cross-sectional evidence on healthy, low-carbon behaviours from UK Biobank

Michaela A Smith,[1] Jan Rasmus Boehnke,[1,2] Hilary Graham,[1] Piran C L White,[3] Stephanie L Prady[1]

[1]Department of Health Sciences, University of York, York, UK
[2]School of Nursing and Health Sciences, University of Dundee, Dundee, UK
[3]Department of Environment and Geography, University of York, York, UK

**Correspondence to**
Dr Michaela A Smith;
michaela.smith@alumni.york.ac.uk

## ABSTRACT

**Objectives** To examine whether there are associations between active travel and markers of a healthy, low-carbon (HLC) diet (increased consumption of fruit and vegetables (FV), reduced consumption of red and processed meat (RPM)).

**Design** Cross-sectional analysis of a cohort study.

**Settings** Population cohort of over 500 000 people recruited from 22 centres across the UK. Participants aged between 40 and 69 years were recruited between 2006 and 2010.

**Participants** 412 299 adults with complete data on travel mode use, consumption of FV and RPM, and sociodemographic covariates were included in the analysis.

**Exposure measures** Mutually exclusive mode or mode combinations of travel (car, public transport, walking, cycling) for non-work and commuting journeys.

**Outcome measures** Consumption of FV measured as portions per day and RPM measured as frequency per week.

**Results** Engaging in all types of active travel was positively associated with higher FV consumption and negatively associated with more frequent RPM consumption. Cycling exclusively or in combination with walking was most strongly associated with increased dietary consumption of FV and reduced consumption of RPM for both non-work and commuting journeys. Overall, the strongest associations were between non-work cycling and FV consumption (males: adjusted OR=2.18, 95% CI 2.06 to 2.30; females: adjusted OR=2.50, 95% CI 2.31 to 2.71) and non-work cycling and RPM consumption (males: adjusted OR=0.57, 95% CI 0.54 to 0.60; females: adjusted OR=0.54, 95% CI 0.50 to 0.59). Associations were generally similar for both commuting and non-work travel, and were robust to adjustment with sociodemographic and behavioural factors.

**Conclusions** There are strong associations between engaging in active travel, particularly cycling, and HLC dietary consumption, suggesting that these HLC behaviours are related. Further research is needed to better understand the drivers and dynamics between these behaviours within individuals, and whether they share common underlying causes.

### Strengths and limitations of this study

► This study uses the UK Biobank to examine associations between active travel and diet in order to better understand the patterning of healthy, low-carbon lifestyles.
► UK Biobank is a large cohort with very rich data, which enabled assessment of relationships using several measures of travel and dietary behaviour and adjustment for a wide range of sociodemographic, environmental and behavioural covariates.
► The analysis used in this study is cross sectional and therefore we cannot infer causality between these behaviours.
► This study is further limited by the use of self-reported measures of active travel and dietary consumption.

## INTRODUCTION

Increasing concerns about global climate change combined with rising rates of chronic disease have led to greater policy attention on behaviours and lifestyles that are beneficial for both human health and the natural environment.[1–3] From this perspective, two priority behaviours that have been identified are engaging in healthy, low-carbon (HLC) travel (eg, walking and cycling for transport) and consuming HLC diets (eg, reduced consumption of meat, increased consumption of fruit and vegetables (FV)).[4–8] Modelling studies have shown that a population shift towards these behaviours would lead to a range of health and environmental cobenefits: increased life expectancies, decreased rates of type 2 diabetes, cardiovascular diseases and cancer, as well as large reductions in transport and food-related greenhouse gas emissions.[3 6 7 9–12] These shifts are also in line with national health guidelines. In the UK, for example, walking and cycling for transport is widely recommended

as a key strategy to increase population physical activity (PA),[13 14] and adults are advised to base two-thirds of their diet on plant-sourced foods, specifically consuming at least five portions of FV and less than 70 g of red and processed meat (RPM) per day to prevent chronic disease outcomes.[15 16] These dietary principles are also in alignment with the recently published 'planetary health diet' which argues that huge changes in consumption of fruits, vegetables and red meat are needed on a global scale if we are to stay within safe planetary boundaries.[17]

In the UK, studies that have examined these travel and dietary behaviours at the population level have found that they are strongly patterned by sociodemographic factors,[18–22] which suggests that HLC behaviours may overlap among certain population groups and/or within specific environments. Nevertheless, it remains unclear whether these behaviours actually co-occur within the same individuals, as there are very few studies that have examined HLC travel and dietary behaviours in relation to each other. For example, evidence from surveys and psychological research has shown that people who are willing to drive less (or drive more efficiently) are also more willing to reduce their meat consumption, but these associations have been limited to behavioural intentions rather than actual travel behaviour and food consumption.[23–25] At the same time, there is considerable evidence of positive associations between PA and consuming more healthful diets,[26–29] but it is not known whether this relationship also extends to forms of physically active *travel* or to diets that are both healthy and low carbon. Based on this evidence, it has been proposed that strategies to promote active travel could also offer additional population health benefits through indirect dietary outcomes,[30] but these relationships are poorly understood and have not yet been tested empirically. Determining whether behaviours co-occur is important because if behaviours are related, engaging in one behaviour may modify the likelihood of engaging in another.[31–34] This means that strategies which target multiple behaviours together may have additional benefits over the sum of individual interventions,[35] and therefore have the potential to produce synergistic outcomes. Indeed, this potential for positive interactions makes it particularly important to tease out relationships between active travel and dietary consumption, as it is possible that the observed associations between walking/cycling and better health outcomes in the literature may be partially attributable to the dietary patterns of active travellers (and/or vice versa).

In light of these gaps, the objective of this study was to explore relationships between HLC behaviours in the travel and dietary domains, by examining associations between engaging in active travel and consumption of two food groups (FV, RPM) that have contrasting implications for human health and carbon emissions. Our choice of measures was based on the behaviours for which there are UK government recommendations and for which there is the greatest evidence of combined public health and environment benefits. As far as we are aware, there has been no prior research that has explicitly examined the relationships between these combinations of diet and travel behaviour.

## METHODS

### Study design and sample

We used baseline data from UK Biobank (UKB; project 14840) to assess cross-sectional relationships between use of different travel modes and dietary consumption. The scientific rationale, study design and survey methods for UKB have been described elsewhere.[36] Briefly, data were collected from 502 616 individuals aged 40–69 years recruited between 2006 and 2010. Participants were identified from National Health Service patient registers and invited to attend 1 of 22 assessment centres located throughout the UK. At each assessment centre, participants completed a touchscreen questionnaire that collected information on sociodemographic characteristics and diet, lifestyle and environmental factors.

In this study, participants who did not provide any information on travel mode use (n=7272) or dietary consumption (FV or RPM, n=1820) were excluded, yielding an initial sample size of 493 524. This number was then further restricted to participants who had complete data on all analytical covariates (n=412 299 for all journeys, n=234 148 for commuting journeys). Sensitivity analyses were conducted with a further subsample that had complete data on weekly PA and total energy intake (95 475 females and 83 213 males).

### Measures

#### Travel mode use

Data on travel behaviour were collected on the touchscreen questionnaire. Participants were asked to report which travel mode(s) they used for non-work journeys (*In the last 4 weeks, which forms of transport have you used most often to get about?*) and for their travel to work (commuting journeys), if they were currently employed and did not always work from home (*What types of transport do you use to get to and from work?*). Both questions had the same response options (car/motor vehicle, public transport, walking, cycling), and allowed participants to select multiple modes for each type of journey.

Using these two questions, we categorised travel behaviour in several ways. First, to create an overall measure of active travel for each participant, we combined the responses from the two travel questions into one binary variable which included those who reported any walking or any cycling for either non-work or commuting journeys. Similar binary variables were also created for any walking and any cycling across the two types of journeys. Second, to account for all possible combinations of travel, a 15-category travel mode variable was derived for each type of journey (non-work, commuting) in order to organise the modal combinations from those producing the most carbon emissions and requiring the least physical exertion (car use only), to those producing the

least emissions and requiring the most physical exertion (cycling only or cycling+walking). This was then collapsed into an eight-category variable for each type of journey: (1) car only, (2) car+public transport only, (3) car+public and active transport, (4) car+active transport only, (5) public transport only, (6) public+active transport, (7) walking only, and (8) cycling only or cycling+walking. This approach is similar to that used previously by Flint and Cummins.[37]

## Dietary consumption

Data on FV and RPM consumption also came from the touchscreen questionnaire. Participants were asked to report their FV consumption via four open-ended questions that asked about the average number of tablespoons of vegetables and pieces of fruit consumed each day. These responses were then recoded into standard '5-a-day' portions[38] that resulted in an overall measure of average portions of FV consumed for each participant. To assess whether each participant's consumption was in line with the recommended guideline, this variable was also recoded into a three-level ordinal measure: <3, 3 to <5 and 5+ portions/day.

Participants were asked five questions about their average weekly intake of different types of meat. To create an overall measure of RPM consumption, we combined the four questions involving RPM (beef, lamb, pork, processed meat) into a composite index, based on the number of times each type of meat was consumed on a weekly basis.[39] For each meat type, the responses were coded as follows: never=0, less than once a week=0.5, once a week=1, two to four times a week=3, five to six times a week=5.5, once or more daily=7. This index variable ranged from 0 to 28, where 0 indicated that participants never consumed any RPM and 28 indicated that participants consumed all four types of RPM on a daily basis. Based on the distribution of the resulting index variable, RPM consumption was then grouped into three categories: (1) non-consumers, and consumers split at the median frequency; (2) up to three times per week; and (3) >3 times per week. This approach was used by Bradbury et al,[39] who showed that those who consume RPM most frequently (>3 times per week) in the UKB sample also consume the largest quantities per day.

## Covariates

Various demographic, socioeconomic and environmental factors were hypothesised as possible confounders to relationships between travel behaviour and dietary consumption. Demographic covariates were age at baseline, sex, ethnic origin and household size. Socioeconomic covariates were gross annual household income, number of cars per household, highest educational qualification and occupational class. We used the National Statistics Socioeconomic Classification for occupation class by converting codes from Standard Occupational Classification 2000. Environmental covariates were residential area classification, Townsend deprivation score and region of UK.

Weekly PA (meeting or not meeting PA guideline) and total energy intake (kcal) were used in sensitivity analyses, due to the complex inter-relationships between active travel, PA, dietary consumption and energy intake (see further details and diagram of putative relationships in online supplementary appendix figure S1). Those who reported 150 min of moderate PA or 75 min of vigorous PA per week were considered to meet the current PA guideline.[40] Data on total energy intake came from a 24-hour dietary recall questionnaire which was completed at the assessment centre by the last 70 000 participants and up to four times by email in the rest of the cohort.[41] For respondents who completed multiple dietary recall questionnaires, we used the average value.

Covariates were mostly self-reported on the touchscreen questionnaire, with the exception of occupational class (verbal interview), residential area classification (census), Townsend deprivation score (census), region of UK (assessment centre location) and average energy intake (24-hour dietary assessment).

## Statistical analysis

Associations between each measure of travel behaviour and each dietary outcome were examined using multivariate ordinal regression models in Stata/SE V.14.0.[42] We used ordinal logistic regression in order to model the trends in dietary consumption while keeping the 'extremes' as useful categories (eg, non-consumers of RPM, and those who met or exceeded consumption guidelines). This enabled meaningful interpretation of the relationships with a view to national dietary recommendations and potentially discontinuous changes in the associations between travel and dietary behaviour. Though these relationships could plausibly go in either direction, we modelled them in this way based on previous hypotheses[30] as well as neurocognitive research which suggests that PA may be more likely to lead to dietary changes than vice versa.[43 44]

In model 1 we examined the bivariate association between each travel variable and each dietary outcome, and in model 2 we adjusted for sociodemographic and environmental covariates. As a sensitivity analysis we further adjusted for PA and energy intake (model 3) in the subsample with complete data on these factors (for comparison purposes models 1 and 2 were rerun in this subsample as well). This sensitivity analysis was only conducted for the any active travel variable since this contained all of the other active travel combinations.

When interpreting the ordinal logistic model, the model assumes that the relationship between each pair of outcome groups is the same, or in other words, that the coefficients describing the relationship between the lowest outcome category and all higher categories are the same as those describing the relationship between the next lowest category and all higher categories, and so on. This is called the proportional odds or parallel

**Table 1** Descriptive characteristics of the sample (n=412 299)

| | Males | | Females | | All | |
|---|---|---|---|---|---|---|
| | n | % | n | % | n | % |
| Total | 195 131 | 47.3 | 217 168 | 52.7 | 412 299 | 100.0 |
| Age at baseline (years) * | | | | | | |
| 38–44 | 20 476 | 10.5 | 23 892 | 11.0 | 44 368 | 10.8 |
| 45–49 | 25 246 | 12.9 | 31 543 | 14.5 | 56 789 | 13.8 |
| 50–54 | 28 821 | 14.8 | 36 394 | 16.8 | 65 215 | 15.8 |
| 55–59 | 34 774 | 17.8 | 40 910 | 18.8 | 75 684 | 18.4 |
| 60–64 | 46 955 | 24.1 | 50 174 | 23.1 | 97 129 | 23.6 |
| 65–73 | 38 859 | 19.9 | 34 255 | 15.8 | 73 114 | 17.7 |
| Ethnic group | | | | | | |
| White British | 175 294 | 89.8 | 193 220 | 89.0 | 368 514 | 89.4 |
| Other White | 10 855 | 5.6 | 13 903 | 6.4 | 24 758 | 6.0 |
| South Asian | 3835 | 2.0 | 2870 | 1.3 | 6705 | 1.6 |
| Black | 2403 | 1.2 | 3306 | 1.5 | 5709 | 1.4 |
| Chinese | 450 | 0.2 | 720 | 0.3 | 1170 | 0.3 |
| Mixed | 891 | 0.5 | 1448 | 0.7 | 2339 | 0.6 |
| Other | 1403 | 0.7 | 1701 | 0.8 | 3104 | 0.8 |
| Highest qualification † | | | | | | |
| College or university degree | 70 136 | 35.9 | 74 613 | 34.4 | 144 749 | 35.1 |
| A levels or equivalent | 20 898 | 10.7 | 27 183 | 12.5 | 48 081 | 11.7 |
| GCSEs or equivalent | 36 862 | 18.9 | 51 055 | 23.5 | 87 917 | 21.3 |
| CSEs or equivalent | 10 560 | 5.4 | 11 730 | 5.4 | 22 290 | 5.4 |
| NVQ or HND or HNC or equivalent | 17 732 | 9.1 | 9607 | 4.4 | 27 339 | 6.6 |
| Other professional qualifications | 8560 | 4.4 | 12 375 | 5.7 | 20 935 | 5.1 |
| No qualifications | 30 383 | 15.6 | 30 605 | 14.1 | 60 988 | 14.8 |
| Occupational class ‡ | | | | | | |
| Higher managerial and professional | 48 981 | 25.1 | 25 058 | 11.5 | 74 039 | 18.0 |
| Lower managerial and professional | 34 686 | 17.8 | 54 458 | 25.1 | 89 144 | 21.6 |
| Intermediate occupations | 14 933 | 7.7 | 36 723 | 16.9 | 51 656 | 12.5 |
| Small employers and own accounts | 9345 | 4.8 | 4958 | 2.3 | 14 303 | 3.5 |
| Lower supervisory and technical | 10 702 | 5.5 | 1019 | 0.5 | 11 721 | 2.8 |
| Semiroutine occupations | 10 986 | 5.6 | 20 181 | 9.3 | 31 167 | 7.6 |
| Routine occupations | 10 162 | 5.2 | 5365 | 2.5 | 15 527 | 3.8 |
| Not classified | 55 336 | 28.4 | 69 406 | 32.0 | 124 742 | 30.3 |
| Household income (before tax) | | | | | | |
| Less than £18 000 | 39 184 | 20.1 | 52 863 | 24.3 | 92 047 | 22.3 |
| £18 000–£30 999 | 47 701 | 24.5 | 57 347 | 26.4 | 105 048 | 25.5 |
| £31 000–£51 999 | 52 674 | 27.0 | 55 578 | 25.6 | 108 252 | 26.3 |
| £52 000–£100 000 | 43 674 | 22.4 | 40 867 | 18.8 | 84 541 | 20.5 |
| Greater than £100 000 | 11 898 | 6.1 | 10 513 | 4.8 | 22 411 | 5.4 |
| Household size | | | | | | |
| 1 | 33 345 | 17.1 | 45 334 | 20.9 | 78 679 | 19.1 |
| 2 | 90 130 | 46.2 | 98 297 | 45.3 | 188 427 | 45.7 |
| 3 | 30 803 | 15.8 | 33 989 | 15.7 | 64 792 | 15.7 |
| 4 | 29 408 | 15.1 | 28 809 | 13.3 | 58 217 | 14.1 |
| 5+ | 11 445 | 5.9 | 10 739 | 4.9 | 22 184 | 5.4 |
| Number of cars per household | | | | | | |

Continued

**Table 1** Continued

|  | Males | | Females | | All | |
|---|---|---|---|---|---|---|
|  | n | % | n | % | n | % |
| 0 | 14 877 | 7.6 | 19 055 | 8.8 | 33 932 | 8.2 |
| 1 | 77 536 | 39.7 | 95 160 | 43.8 | 172 696 | 41.9 |
| 2 | 79 161 | 40.6 | 79 599 | 36.7 | 158 760 | 38.5 |
| 3 | 17 829 | 9.1 | 17 994 | 8.3 | 35 823 | 8.7 |
| 4+ | 5728 | 2.9 | 5360 | 2.5 | 11 088 | 2.7 |
| Region§ |  |  |  |  |  |  |
| London | 25 333 | 13.0 | 30 273 | 13.9 | 55 606 | 13.5 |
| South-East England | 17 007 | 8.7 | 19 402 | 8.9 | 36 409 | 8.8 |
| South-West England | 16 764 | 8.6 | 19 613 | 9.0 | 36 377 | 8.8 |
| East Midlands | 13 120 | 6.7 | 14 559 | 6.7 | 27 679 | 6.7 |
| West Midlands | 18 383 | 9.4 | 18 020 | 8.3 | 36 403 | 8.8 |
| Yorkshire and the Humber | 29 615 | 15.2 | 32 479 | 15.0 | 62 094 | 15.1 |
| North-East England | 23 110 | 11.8 | 25 606 | 11.8 | 48 716 | 11.8 |
| North-West England | 29 599 | 15.2 | 31 717 | 14.6 | 61 316 | 14.9 |
| Wales | 8265 | 4.2 | 9048 | 4.2 | 17 313 | 4.2 |
| Scotland | 13 935 | 7.1 | 16 451 | 7.6 | 30 386 | 7.4 |
| Urban residence | 167 547 | 85.9 | 186 617 | 85.9 | 354 164 | 85.9 |
| Townsend score (mean, SD)¶ | −1.37 | 3.1 | −1.37 | 3.0 | −1.37 | 3.0 |

*Continuous variable in models
†A levels: academic advanced levels, postcompulsory education. GCSE: academic General Certificate of Secondary Education, formerly ordinary levels, taken at ages 15–16 years and the end of compulsory education. CSE: vocational Certificate of Secondary Education, formerly taken at ages 15–16 years. NVQ, HND, HNC: National Vocational Qualifications, Higher National Diploma, Higher National Certificate, all intermediate semivocational qualifications.
‡Based on National Statistics Socio-economic Classification (NS-SEC), where not classified=those who were retired, unemployed, looking after home/family, unable to work because of sickness/disability, doing unpaid/voluntary work, or full-time students.
§Grouped based on assessment centre: London=St Barts, Croydon, Hounslow; South-East England=Oxford, Reading; South-West England=Bristol; East Midlands=Nottingham; West Midlands=Birmingham; Yorkshire and the Humber=Leeds, Sheffield; North-East England=Middlesbrough, Newcastle; North-West England=Liverpool, Manchester, Bury; Wales=Cardiff, Swansea, Wrexham; Scotland=Glasgow, Edinburgh.
¶Lower score=less deprived (min: −6.3; max: 11.0).

lines assumption,[45] and in this case, the models assume that the odds of being in the lowest dietary consumption category compared with the two highest, are the same as the odds of being in the highest consumption category compared with the two lowest. In each regression model, we tested the proportional odds assumption using the Stata *oparallel* postestimation command.[46] Where this assumption was not met (p<0.05), we reran each model as a generalised ordered logit model (Stata extension *gologit2*) which relaxes the proportional odds assumption for some predictor variables while maintaining it for others.[47] This approach has the advantages of being more parsimonious and interpretable than those estimated by a non-ordinal method and may also give added insights (eg, discontinuous changes) into the data that would be lost by ignoring the differences and continuing to use the fully ordinal model.[45] We present ORs or adjusted ORs (aOR) with 95% CIs and set a threshold of alpha=0.05 for statistical significance. All analyses were stratified by sex due to established gender differences in the patterning of travel behaviour and dietary consumption in the UK population.[19 21 48 49]

### Patient and public involvement

This study was conducted using the UKB resource. Details of patient and public involvement in the UKB are available online (https://www.ukbiobank.ac.uk/public-consultation/). No patients were specifically involved in setting the research question or the outcome measures, nor were they involved in developing plans for recruitment, design or implementation of this study. No patients were asked to advise on interpretation or writing up of results. There are no specific plans to disseminate the results of the research to study participants, but the UKB disseminates key findings from projects on its website (https://www.ukbiobank.ac.uk/participant-events/).

### RESULTS

Descriptive characteristics of the sample are presented in table 1. As well as being older, UKB participants are more socioeconomically advantaged, and more health conscious in comparison with the UK general population.[50] In this study, 54.5% of the sample reported walking or cycling for either type of journey (any active travel),

**Table 2** Descriptive overview of travel mode use (n=412 299)

| | Males (n=195 131) | | Females (n=217 168) | | All (n=412 299) | |
|---|---|---|---|---|---|---|
| | n | % | n | % | n | % |
| Any active travel * | 105 287 | 54.0 | 119 244 | 54.9 | 224 531 | 54.5 |
| Any walking travel | 96 976 | 49.7 | 115 573 | 53.2 | 212 549 | 51.6 |
| Any cycling travel | 24 806 | 12.7 | 13,877 | 6.4 | 38 683 | 9.4 |
| Non-work journeys † | | | | | | |
| Car only | 79 582 | 40.9 | 82,980 | 38.3 | 162 562 | 39.5 |
| Car+PT | 6058 | 3.1 | 8822 | 4.1 | 14 880 | 3.6 |
| Car+mixed (PT and AT) | 25 683 | 13.2 | 32 024 | 14.8 | 57 707 | 14.0 |
| Car+AT | 44 488 | 22.8 | 46 117 | 21.3 | 90 605 | 22.0 |
| PT only | 9957 | 5.1 | 13 277 | 6.1 | 23 234 | 5.6 |
| PT+AT | 11 793 | 6.1 | 16 020 | 7.4 | 27 813 | 6.8 |
| Walking only | 12 553 | 6.5 | 14 939 | 6.9 | 27 492 | 6.7 |
| Cycling/cycling+walking | 4660 | 2.4 | 2648 | 1.2 | 7308 | 1.8 |
| Missing | 357 | | 341 | | 698 | |
| Commuting journeys ‡ | | | | | | |
| Car only | 74 043 | 65.5 | 73 736 | 60.9 | 147 779 | 63.1 |
| Car+PT | 6735 | 6.0 | 7519 | 6.2 | 14 254 | 6.1 |
| Car+mixed (PT and AT) | 3649 | 3.2 | 3578 | 3.0 | 7227 | 3.1 |
| Car+AT | 7573 | 6.7 | 8727 | 7.2 | 16 300 | 7.0 |
| PT only | 8383 | 7.4 | 12 042 | 10.0 | 20 425 | 8.7 |
| PT+AT | 4861 | 4.3 | 5081 | 4.2 | 9942 | 4.3 |
| Walking only | 3878 | 3.4 | 8183 | 6.8 | 12 061 | 5.2 |
| Cycling/cycling+walking | 3964 | 3.5 | 2196 | 1.8 | 6160 | 2.6 |
| Missing/not applicable | 82 045 | | 96 106 | | 178 151 | |

*Includes walking or cycling for non-work or commuting travel.
†n=411 601 for non-work travel (0.2% missing), % are calculated from non-missing.
‡n=234 148 for commuting travel (43.2% missing/not applicable), % are calculated from non-missing.
AT, active travel; PT, public transport.

and walking was much more common than cycling (51.6% vs 9.4%) (table 2). Car only travel was much higher for commuting journeys (63.1%) than for non-work journeys (39.5%), indicating that people were more likely to use multiple modes and more active modes for non-work travel. For example, 22.0% of the sample mixed car use with active modes, and 14.0% mixed car, active modes and public transport use for non-work journeys. For diet, 58.3% of males and 36.7% of females reported consuming RPM more than three times per week and only 5.3% of the sample reported never consuming any RPM (3.4% among males, 7.0% among females) (table 3). Nearly 38% reported consuming 5+ portions of FV per day on average (31.4% among males, 43.3% among females).

### Associations between travel modes and FV consumption
Across all models, there were positive associations between all types of HLC travel and FV consumption among both males and females, with very little change even after adjustment for demographic, socioeconomic and environmental factors (table 4 and online supplementary appendix figure S2). Associations were generally much stronger for cycling than for other travel modes. For example, in the fully adjusted models (model 2), men and women who engaged in any cycling travel were nearly twice as likely to consume higher amounts of FV than those who did not cycle for transport (males: aOR=1.65, 95% CI 1.61 to 1.69; females: aOR=1.67, 95% CI 1.62 to 1.73). Across the more detailed travel classifications of non-work and commuting journeys, associations were generally weaker or non-significant for travel that did not involve any walking or cycling (eg, car+public transport, public transport only). Comparing across the two types of journeys, the associations were fairly similar in magnitude, though they were slightly stronger for non-work travel, and particularly for non-work cycling. Based on the CIs, women who engaged in any active travel (aOR=1.43, 95% CI 1.40 to 1.45) or any walking travel (aOR=1.38, 95% CI 1.36 to 1.41) were more likely to consume higher

**Table 3** Descriptive overview of dietary consumption and physical activity (n=412 299)

| | Males (n=195 131) | | Females (n=217 168) | | All (n=412 299) | |
|---|---|---|---|---|---|---|
| | n | % | n | % | n | % |
| **FV consumption (portions/day)** | | | | | | |
| <3 | 66 672 | 34.2 | 45 669 | 21.0 | 112 341 | 27.3 |
| 3 to <5 | 67 263 | 34.5 | 77 583 | 35.7 | 144 846 | 35.1 |
| 5+ | 61 196 | 31.4 | 93 917 | 43.3 | 155 112 | 37.6 |
| **RPM consumption (frequency/week)** | | | | | | |
| Never | 6615 | 3.4 | 15 250 | 7.0 | 21 865 | 5.3 |
| ≤3 times | 74 766 | 38.3 | 122 148 | 56.3 | 196 914 | 47.8 |
| >3 times | 113 750 | 58.3 | 79 770 | 36.7 | 193 520 | 46.9 |
| **Meets physical activity guideline*** | | | | | | |
| Yes | 101 323 | 54.1 | 103 804 | 50.0 | 205 127 | 52.0 |
| No | 86 112 | 45.9 | 103 996 | 50.0 | 189 108 | 48.0 |
| Missing | 7696 | | 10 368 | | 18 064 | |
| Total energy intake, kcal/day (mean, SD) † | 2299 | 685 | 1971 | 575 | 2123 | 649 |

*n=394 235 for physical activity guideline (4.4% missing), % are calculated from non-missing.
†Based on n=98 853 females, n=85 392 males.
FV, fruit and vegetables; RPM, red and processed meat.

amounts of FV compared with males (aOR=1.35, 95% CI 1.33 to 1.37 and aOR=1.25, 95% CI 1.23 to 1.27, respectively). For non-work journeys, the same was also true for women who used car+active travel, walking only and cycling/cycling+walking, compared with their male counterparts. Full models are shown in online supplementary appendix tables S1 and S2.

### Associations between travel modes and RPM consumption

Overall, the associations between HLC travel and RPM consumption were nearly all negative; the only exception was for car+public transport (vs car only travel) among females for non-work journeys (table 5 and online supplementary appendix figure S3). Among both males and females, associations were only slightly attenuated with adjustment for demographic, socioeconomic and environmental factors. As with FV consumption, these associations were strongest for cycling, overall and across both types of journeys. Moreover, there was a clear gradient of effect for non-work travel, such that the more active the travel mode(s), the more negative the association with RPM consumption frequency. For example, in the fully adjusted models (model 2), men and women who cycled for non-work journeys were nearly half as likely to consume RPM more frequently than those who travelled by car (males: aOR=0.57; 95% CI 0.54 to 0.60; females: aOR=0.54, 95% CI 0.50 to 0.59). Full models are shown in online supplementary appendix tables S3 and S4.

### Proportional odds assumption

Due to the very large sample size in UKB, we were able to detect very minor variations in the data, and this meant that all of the models in tables 4 and 5 had violations of

the proportional odds assumption. To assess whether these differences were meaningful for the key variables of interest (travel variables), all of the models were rerun using a generalised ordered logit model (online supplementary appendix tables S5 and S6). Here the associations were generally of similar magnitude and in the same direction to the fully ordinal models, but where differences were present, the associations tended to be slightly stronger for the two highest categories versus the lowest category of the outcome variables, for example, 3+ portions of FV versus <3, and RPM consumers versus never consumers. This relatively trivial difference does not alter the directions of the associations (positive and negative) in our main findings.

### Sensitivity analyses

In the subset of the sample with full data on energy intake and PA (n=95 475 females, n=83 213 males), adjusting for these variables in addition to the other sociodemographic and environmental factors slightly attenuated the associations between any active travel and FV consumption, but the relationship was still independent and highly significant among both males and females (males: aOR=1.28; 95% CI 1.24 to 1.31 and females: aOR=1.35, 95% CI 1.32 to 1.39) (online supplementary appendix tables S7 and S8). Similarly, the associations between any active travel and RPM consumption were also very slightly attenuated, but even less so than for FV consumption (males: aOR=0.89; 95% CI 0.87 to 0.92 and females: aOR=0.90, 95% CI 0.88 to 0.92) (online supplementary appendix tables S9 and S10).

**Table 4** Ordinal logistic models between HLC travel and FV consumption, stratified by gender (n=412 299)

| Travel variables | Males (n=195 131) | | Females (n=217 168) | |
|---|---|---|---|---|
| | Model 1† | Model 2‡ | Model 1† | Model 2‡ |
| | OR (95% CI) | | | |
| Any active travel (ref: none) | 1.37*** | 1.35*** | 1.42*** | 1.43*** |
| | (1.34 to 1.39) | (1.33 to 1.37) | (1.40 to 1.44) | (1.40 to 1.45) |
| Any walking (ref: none) | 1.28*** | 1.25*** | 1.38*** | 1.38*** |
| | (1.26 to 1.31) | (1.23 to 1.27) | (1.36 to 1.40) | (1.36 to 1.41) |
| Any cycling (ref: none) | 1.57*** | 1.65*** | 1.58*** | 1.67*** |
| | (1.54 to 1.61) | (1.61 to 1.69) | (1.53 to 1.63) | (1.62 to 1.73) |
| Non-work travel§ (ref: car only) | | | | |
| Car+public transport | 1.06* | 1.00 | 1.05* | 0.98 |
| | (1.01 to 1.11) | (0.95 to 1.05) | (1.01 to 1.09) | (0.94 to 1.02) |
| Car+mixed (public and active) | 1.49*** | 1.37*** | 1.57*** | 1.41*** |
| | (1.46 to 1.53) | (1.33 to 1.40) | (1.53 to 1.61) | (1.38 to 1.45) |
| Car+active travel | 1.27*** | 1.26*** | 1.37*** | 1.39*** |
| | (1.24 to 1.29) | (1.24 to 1.29) | (1.34 to 1.40) | (1.36 to 1.42) |
| Public transport only | 1.03 | 1.13*** | 1.03 | 1.11*** |
| | (0.99 to 1.07) | (1.08 to 1.18) | (0.99 to 1.06) | (1.06 to 1.15) |
| Public transport+active travel | 1.31*** | 1.43*** | 1.45*** | 1.52*** |
| | (1.27 to 1.36) | (1.37 to 1.49) | (1.40 to 1.50) | (1.47 to 1.58) |
| Walking only | 1.34*** | 1.39*** | 1.47*** | 1.57*** |
| | (1.29 to 1.38) | (1.34 to 1.44) | (1.42 to 1.52) | (1.51 to 1.62) |
| Cycling/cycling+walking | 2.06*** | 2.18*** | 2.34*** | 2.50*** |
| | (1.95 to 2.17) | (2.06 to 2.30) | (2.17 to 2.52) | (2.31 to 2.71) |
| Commuting travel¶ (ref: car only) | | | | |
| Car+public transport | 1.03 | 0.97 | 1.02 | 0.99 |
| | (0.98 to 1.08) | (0.92 to 1.01) | (0.97 to 1.06) | (0.95 to 1.03) |
| Car+mixed (public and active) | 1.44*** | 1.37*** | 1.45*** | 1.41*** |
| | (1.36 to 1.53) | (1.29 to 1.46) | (1.36 to 1.54) | (1.32 to 1.50) |
| Car+active travel | 1.41*** | 1.47*** | 1.23*** | 1.30*** |
| | (1.35 to 1.47) | (1.41 to 1.54) | (1.18 to 1.28) | (1.24 to 1.35) |
| Public transport only | 1.08*** | 1.03 | 0.91*** | 0.95* |
| | (1.03 to 1.12) | (0.98 to 1.08) | (0.88 to 0.95) | (0.91 to 0.99) |
| Public transport+active travel | 1.41*** | 1.37*** | 1.26*** | 1.28*** |
| | (1.33 to 1.48) | (1.29 to 1.45) | (1.19 to 1.32) | (1.20 to 1.35) |
| Walking only | 1.20*** | 1.24*** | 1.07*** | 1.20*** |
| | (1.13 to 1.28) | (1.16 to 1.32) | (1.03 to 1.12) | (1.14 to 1.25) |
| Cycling/cycling+walking | 1.78*** | 1.82*** | 1.93*** | 2.00*** |
| | (1.68 to 1.89) | (1.71 to 1.93) | (1.77 to 2.09) | (1.84 to 2.18) |

*p<0.05; **p<0.01; ***p<0.001.
†Model 1: unadjusted.
‡Model 2: adjusted for age, ethnic group, education, occupational class, household income, household size, number of cars, assessment centre location, population density, Townsend score (see full models in online supplementary appendix).
§n=194 774 males, n=216 827 females.
¶n=113 086 males, n=121 062 females.
FV, fruit and vegetables; HLC, healthy, low carbon.

## DISCUSSION

To our knowledge, this is the first analysis to explicitly examine the relationships between engaging in active travel and HLC dietary consumption, thus beginning to clarify the patterning of HLC lifestyles. We have shown that engaging in active travel, and in particular cycling, is associated with increased consumption of FV and with reduced consumption of RPM in the UKB sample. These

**Table 5** Ordinal logistic models between HLC travel and RPM consumption, stratified by gender (n=412 299)

| | Males (n=195 131) | | Females (n=217 168) | |
| --- | --- | --- | --- | --- |
| | Model 1† | Model 2‡ | Model 1† | Model 2‡ |
| Travel variables | OR (95% CI) | | | |
| Any active travel (ref: none) | 0.87*** | 0.89*** | 0.85*** | 0.88*** |
| | (0.85 to 0.88) | (0.87 to 0.91) | (0.84 to 0.87) | (0.87 to 0.90) |
| Any walking (ref: none) | 0.91*** | 0.94*** | 0.88*** | 0.91*** |
| | (0.89 to 0.93) | (0.92 to 0.95) | (0.86 to 0.89) | (0.89 to 0.92) |
| Any cycling (ref: none) | 0.75*** | 0.76*** | 0.67*** | 0.72*** |
| | (0.73 to 0.77) | (0.74 to 0.78) | (0.65 to 0.69) | (0.69 to 0.74) |
| Non-work travel§ (ref: car only) | | | | |
| Car+public transport | 0.99 | 1.01 | 1.12*** | 1.09*** |
| | (0.94 to 1.04) | (0.95 to 1.06) | (1.07 to 1.17) | (1.04 to 1.14) |
| Car+mixed (public and active) | 0.92*** | 0.96* | 0.93*** | 0.95*** |
| | (0.89 to 0.94) | (0.94 to 0.99) | (0.91 to 0.96) | (0.93 to 0.98) |
| Car+active travel | 0.95*** | 0.96*** | 0.94*** | 0.94*** |
| | (0.93 to 0.98) | (0.94 to 0.98) | (0.92 to 0.96) | (0.92 to 0.97) |
| Public transport only | 0.91*** | 0.89*** | 0.87*** | 0.88*** |
| | (0.87 to 0.95) | (0.85 to 0.94) | (0.84 to 0.90) | (0.84 to 0.91) |
| Public transport+active travel | 0.78*** | 0.77*** | 0.71*** | 0.76*** |
| | (0.75 to 0.81) | (0.74 to 0.81) | (0.69 to 0.74) | (0.73 to 0.79) |
| Walking only | 0.76*** | 0.75*** | 0.70*** | 0.71*** |
| | (0.73 to 0.78) | (0.72 to 0.78) | (0.68 to 0.72) | (0.69 to 0.74) |
| Cycling/cycling+walking | 0.56*** | 0.57*** | 0.50*** | 0.54*** |
| | (0.53 to 0.59) | (0.54 to 0.60) | (0.46 to 0.54) | (0.50 to 0.59) |
| Commuting travel¶ (ref: car only) | | | | |
| Car+public transport | 0.94** | 1.00 | 1.00 | 1.04 |
| | (0.89 to 0.98) | (0.95 to 1.06) | (0.96 to 1.05) | (0.99 to 1.09) |
| Car+mixed (public and active) | 0.82*** | 0.89** | 0.83*** | 0.93* |
| | (0.76 to 0.87) | (0.84 to 0.96) | (0.78 to 0.89) | (0.86 to 0.99) |
| Car+active travel | 0.82*** | 0.83*** | 0.92*** | 0.89*** |
| | (0.78 to 0.86) | (0.79 to 0.87) | (0.88 to 0.96) | (0.85 to 0.93) |
| Public transport only | 0.86*** | 0.95 | 0.89*** | 0.97 |
| | (0.82 to 0.90) | (0.91 to 1.01) | (0.85 to 0.92) | (0.93 to 1.01) |
| Public transport+active travel | 0.70*** | 0.79*** | 0.71*** | 0.84*** |
| | (0.66 to 0.74) | (0.74 to 0.84) | (0.67 to 0.75) | (0.79 to 0.89) |
| Walking only | 0.76*** | 0.80*** | 0.89*** | 0.86*** |
| | (0.71 to 0.81) | (0.75 to 0.86) | (0.85 to 0.93) | (0.82 to 0.91) |
| Cycling/cycling+walking | 0.58*** | 0.60*** | 0.51*** | 0.55*** |
| | (0.54 to 0.62) | (0.56 to 0.64) | (0.46 to 0.55) | (0.50 to 0.60) |

*p<0.05; ** p<0.01; ***p<0.001.
†Model 1: unadjusted.
‡Model 2: adjusted for age, ethnic group, education, occupational class, household income, household size, number of cars, assessment centre location, population density, Townsend score (see full models in online supplementary appendix).
§n=194 774 males, n=216 827 females.
¶n=113 086 males, n=121 062 females.
HLC, healthy, low carbon; RPM, red and processed meat.

associations were robust to adjustment by both socio-demographic and behavioural factors, suggesting that these factors do not explain the observed relationships. Using multiple measures of travel and dietary behaviour, we have assessed these relationships comprehensively across different travel modes, types of journeys and relevant food groups, and also adjusted for a wide range of important covariates. This level of detail has allowed us

to isolate and elucidate where the relationships between these HLC behaviours are strongest and weakest, which is an important contribution to understanding which elements of travel and dietary behaviour may share common underlying factors.

The major strength of this study is the large sample size and flexible measures of travel behaviour in the UKB data set, both of which enabled the observation of relatively fine-grained differences in the data. Nevertheless, UKB is limited by its lack of representativeness, as it is based on a sample of 'healthy volunteers'[50] and excludes large segments of the population (eg, those under age 40). The data were also collected between 2006 and 2010, and there have been some population changes in meat consumption since then, though less so among those in the UKB age range.[51] As a result, it is unclear whether these results are generalisable to the UK general population, however similar relationships were also found when this analysis was replicated in a nationally representative UK sample,[52 53] which supports the external validity of these associations.

Other limitations include that the measures used were all self-reported and that the analyses are cross sectional. Due to the health-conscious nature of the cohort, it is possible that consumption of some food groups may be over-reported or under-reported; however, an in-depth study of the reliability of the UKB touchscreen dietary questionnaire has shown that participant responses for FV and meat consumption are very consistent over time (70%–90%) and correlate well with other independent dietary assessments (eg, 24-hour dietary recall) conducted as part of the larger UKB study.[54] Nonetheless, if participants were more likely to report that they walked, cycled, ate more FV and ate less RPM, then this might partially explain the observed associations between these behaviours. The cross-sectional nature of the data means we cannot establish causality between these behaviours in terms of whether active travel precedes higher FV and lower RPM consumption, vice versa, or whether change occurs in tandem, or when in the life course such patterns emerge or change. Future research with longitudinal data will help confirm the direction of these relationships, as well as improve our understanding of behaviour dynamics over time.

Importantly, the findings of this study confirm much of the wider evidence on links between health and environmental behaviours, and represent some of the strongest evidence to date on this topic. Several studies have reported clustering between increased PA and more nutritious diets,[26–29 55] but this study is the first to show that comparable associations exist for *physically active travel* and healthy diets, independent of overall PA. Our findings also build on studies of environmental behaviours which have linked reduced car driving with reduced meat consumption, but which have only measured behavioural intentions.[24 25] More indirectly, there are also interesting parallels between this study and the growing body of evidence relating active travel, and particularly active commuting, to positive health outcomes like lower obesity and reduced mortality.[19 37 56–59] Two of these studies, also conducted using UKB but only examining active commuting, have found particularly strong effects for cycling to work and lower obesity[37] and reduced mortality,[58] far over and above the effects found for walking. Combining these findings with our results on the dietary patterns of cyclists suggests that positive interactions between cycling travel and HLC diets could be one factor contributing to the enhanced health effects observed among individuals who cycle.

This study has several important implications. First, the results suggest that active travel and HLC diets may be related and share similar determinants within individuals. Theoretical understandings of behavioural co-occurrence suggest that behaviours which cluster together share common causal pathways,[31 32] and that the stronger the relationship between two behaviours, the more determinants they are likely to share.[60] In this study, strong relationships were seen most clearly between cycling and FV consumption, even after adjusting for sociodemographic characteristics and behavioural factors like overall PA and energy intake. This suggests that these behaviours may be driven by common underlying factors, and supports the interpretation of both behaviours being related to health motivations, though there may also be other factors at play. Since cycling is still a relatively rare form of travel in the UK, these patterns may reflect the fact that people who cycle for transport are somewhat unique, and may also deviate from social norms in other ways (eg, diet). Future research could explore this area further by examining whether relationships between cycling and dietary consumption are consistent in parts of the UK where people cycle at higher frequencies (eg, Cambridge)[61] or among those who cycle at higher intensities, such as for sport.

Identifying whether two behaviours are related is important because strongly associated behaviours may influence each other in different ways.[31–34] In the case of positive relationships, this could mean that related behaviours have the potential to produce synergistic outcomes, if strategies that target multiple HLC behaviours together have greater benefits than the sum of individual interventions.[35] Urgent changes in lifestyles are needed if we are to avoid catastrophic climate change.[62 63] Putting these changes into action requires that we have a complete understanding of people's behaviour patterns, including how different behaviours influence, interact and intersect with one another across the life course. Though relationships between active travel and diet still need to be examined longitudinally, this study suggests that these HLC behaviours may have the potential to positively influence one another, and that promotion of these behaviours could help foster enhanced benefits for both human health and the natural environment.

**Acknowledgements** This research has been conducted using the UK Biobank Resource under Application No 14840.

**Contributors** MAS, JRB, HG, PCLW and SLP made substantial contributions to the conception and design of the study and interpretation of data. MAS undertook the statistical analysis with input from JRB and SLP. MAS drafted the article and JRB, HG, PCLW and SLP revised it critically for important intellectual content. MAS, JRB, HG, PCLW and SLP approved the final version of the manuscript to be published.

**Funding** MAS was supported by a PhD studentship from the University of York as part of the Health of Populations and Ecosystems (HOPE) project, funded by the Economic and Social Research Council (grant number ES/L003015/1), awarded to HG and PCLW.

**Competing interests** None declared.

**Ethics approval** UK Biobank received ethics approval from the National Information Governance Board for Health and Social Care and the National Health Service North West Centre for Research Ethics Committee (Ref: 11/NW/0382).

**Provenance and peer review** Not commissioned; externally peer reviewed.

**Data availability statement** This study used data from the UK Biobank (application 14840) which does not permit public sharing of the data. The data are, however, open to all qualified researchers anywhere in the world and can be accessed by applying through the UK Biobank Access Management System (www.ukbiobank.ac.uk/register-apply).

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
