## [Reviewer comments · BMJ Open]

ARTICLE DETAILS

TITLE (PROVISIONAL)	Associations between active travel and diet: cross-sectional evidence on healthy, low-carbon behaviours from UK Biobank
AUTHORS	Smith, Michaela; Boehnke, Jan Rasmus; Graham, Hilary; White, Piran; Prady, Stephanie

VERSION 1 – REVIEW

REVIEWER	Alistair Woodward University of Auckland New Zealand
REVIEW RETURNED	08-May-2019

GENERAL COMMENTS	Very nicely written, this was a pleasure to read. The topic is important, the aims of the paper are well set out, the conclusions are appropriate, in my view. I offer some suggestions and questions that are in the class of 'minor revisions'. The significance of correlations between healthy diets and active transport extend further than described in the introduction – the observed associations between walking and cycling and better health outcomes may be partially explained by diet (and/or vice versa), especially if these relationships are not explained by socio-economic factors of the kind that are conventionally controlled in epidemiological studies. Is this worth further comment? Note also the carbon saving aspect of active travel may be influenced by dietary choices, if for instance energy intake is much increased amongst those who cycle thereby incurring an emissions penalty due to food-related emissions. Did the investigators examine relations between caloric intake and transport behaviours? This area is complicated. A DAG style diagram that displays the putative relationships between diet, physical activity and co-variates, including direction, feedbacks and modifying factors would be helpful. What is the likely extent of reporting bias in the dietary questionnaire? Are there other, non-dietary variables, also subject to socially desirable responses (tobacco use perhaps?) that might shed light on the likely under-reporting of RPM and over-reporting of FV? Hypothesis: in a heavily car-dominated transport environment, those who cycle are likely to deviate from social norms in other respects (eg smoking behavior, dietary choices). But self-selection of this kind is less marked when the prevalence of cycling is high.
--

	Was it possible to stratify by region, comparing the associations between active transport and diet in high-cycling and low-cycling areas of the UK? Was it possible to distinguish between sports cyclists and others who ride bicycles, at less intense levels? Particularly for non-work journeys, this may be relevant.
--	---

REVIEWER	Freya MacMillan Western Sydney University, Australia
REVIEW RETURNED	09-May-2019

GENERAL COMMENTS	This is a very interesting and well written manuscript, exploring the associations between active travel and a healthy, low carbon diet. Novel data is presented indicating an association between active travel participation, especially cycling, and increased consumption of fruit and vegetables and reduced consumption of red and processed meat. The abstract was heavily weighted to the results section but I think this is appropriate for such an article that is drawing on already existing data that has had the design, setting and participants described in detail and published elsewhere.
---

REVIEWER	Martin Berg Johansen Aalborg University Hospital Denmark
REVIEW RETURNED	07-Jun-2019

GENERAL COMMENTS	Dear authors, As a statistical reviewer of your manuscript I would like to congratulate you on a job well done. You have performed an analysis of a rather complex dataset with categorical measures as both outcomes and exposures. This calls for a rather advanced analytical approach and, in my judgement, you have successfully applied the appropriate methods using an ordinal logistic regression model. There are some decisions to be made in such a situation, e.g. the proportional odds assumption, and you have described these and provided arguments and reasoning for each decision and addressed potential violations. The results from the analyses are presented in great detail in tables and summarized in a sensible way in the text and conclusions. You also show due care not to infer any causality from the results. As such, I have no real reservations about the statistical methods presented in your manuscript. The chosen statistical model might be unfamiliar to some readers and they might benefit from an added sentence that describes exactly how to interpret the reported odds ratios from a proportional odds ordered logistic regression model.
---

VERSION 1 – AUTHOR RESPONSE

REVIEWER 1

Comment #1: Very nicely written, this was a pleasure to read. The topic is important, the aims of the paper are well set out, the conclusions are appropriate, in my view. I offer some suggestions and questions that are in the class of 'minor revisions'.

Author response: Thank you for reviewing our paper and for your positive feedback. We appreciate the opportunity to address these minor revisions to our manuscript.

Comment #2: The significance of correlations between healthy diets and active transport extend further than described in the introduction – the observed associations between walking and cycling and better health outcomes may be partially explained by diet (and/or vice versa), especially if these relationships are not explained by socio-economic factors of the kind that are conventionally controlled in epidemiological studies. Is this worth further comment?

Author response: We thank the reviewer for recognizing this point and we agree with its importance. We have already touched on the significance of these correlations in relation to health outcomes in the discussion section (pg 15, lines 7 to 14) and we have also added a sentence to the introduction section of the revised manuscript which elaborates on this further (pg 5, lines 11 to 14):

“Indeed, this potential for positive interactions makes it particularly important to tease out relationships between active travel and dietary consumption, as it is possible that the observed associations between walking/cycling and better health outcomes in the literature may be partially attributable to the dietary patterns of active travellers (and/or vice versa).”

Comment #3: Note also the carbon saving aspect of active travel may be influenced by dietary choices, if for instance energy intake is much increased amongst those who cycle thereby incurring an emissions penalty due to food-related emissions. Did the investigators examine relations between caloric intake and transport behaviours?

Author response: We thank the reviewer for raising this point – we agree that it is important to examine the role of caloric intake, as those who are more physically active may consume more food energy (calories). To account for this, we adjusted for energy intake in our models as a sensitivity analysis (Supplementary Appendix, Tables S7-S10, Model 3) since we only had caloric data for a subset of the cohort. After adjustment, we still found independent relationships between active travel and both dietary behaviours, though the associations were slightly attenuated. This is noted in the paper on page 13, lines 7 to 15, where we have now added more clear references to the tables in the Supplementary Appendix:

“In the subset of the sample with full data on energy intake and physical activity (n=95,475 females, n=83,213 males), adjusting for these variables in addition to the other socio-demographic and environmental factors slightly attenuated the associations between any active travel and FV consumption, but the relationship was still independent and highly significant among both males and females (males: aOR=1.28; 95%CI 1.24, 1.31 and females: aOR=1.35, 95%CI 1.32, 1.39) (Supplementary Appendix, Tables S7 and S8). Similarly, the associations between any active travel and RPM consumption were also very slightly attenuated, but even less so than for FV consumption (males: aOR= 0.89; 95%CI 0.87, 0.92 and females: aOR=0.90, 95%CI 0.88, 0.92) (Supplementary Appendix, Tables S9 and S10).”

Comment #4: This area is complicated. A DAG style diagram that displays the putative relationships between diet, physical activity and co-variates, including direction, feedbacks and modifying factors would be helpful.

Author response: We thank the reviewer for this suggestion, and we agree that a diagram displaying putative relationships between these interrelated variables would be a helpful addition to the paper. We note however that our study was not planned and conducted using a fully DAG informed approach and retro-fitting this would have necessitated a major re-write discussing the individual factors and their causal inter-relationships. As presented in the manuscript, we followed a general

approach of controlling for previously identified relationships of our two target behaviours (active travel and dietary consumption). Since journal guidelines limit the paper to 5 tables/figures, we have added a new diagram to the first page of the Supplementary Appendix that depicts these relationships (Figure S1) and made reference to this in the revised manuscript in the covariates section (pg 8, lines 22 to 24):

“Weekly PA (meeting or not meeting PA guideline) and total energy intake (kcal) were used in sensitivity analyses, due to the complex interrelationships between active travel, physical activity, dietary consumption and energy intake (further details and diagram of putative relationships in Supplementary Appendix, Figure S1).”

In addition, we have added this explanation to the Supplementary Appendix on page 2 to accompany our added diagram:

“Figure S1 provides an overview of the putative relationships between active travel, dietary consumption, physical activity, energy intake and other covariates in this study.

People who are more physically active tend to have healthier diets [1] and may be more likely to engage in active travel [2] – this means that one’s initial physical activity level (time 1) may be a confounder of the relationship between active travel and dietary consumption, in the same way that demographic, socio-economic, and environmental factors may also act as confounders.

At the same time, we also know that those who engage in active travel may accumulate additional physical activity [3], which may ultimately lead them to consume more food and have a higher energy intake (time 2). In this way, it is possible for physical activity and energy intake to act as mediators of the relationship between active travel and dietary consumption.

It is not possible to tease out these distinctions with cross-sectional data, where all of these variables have been measured at the same point in time. To account for this limitation, we have presented three models with different levels of covariate adjustment: Model 1 is unadjusted, Model 2 is adjusted for demographic, socio-economic and environmental factors, and Model 3 (sensitivity analysis) is adjusted for physical activity level and energy intake.”

Comment #5: What is the likely extent of reporting bias in the dietary questionnaire? Are there other, non-dietary variables, also subject to socially desirable responses (tobacco use perhaps?) that might shed light on the likely under-reporting of RPM and over-reporting of FV?

Author response: We thank the reviewer for raising this important question. Previously, several studies have conducted an in-depth examination into the representativeness of the UK Biobank cohort and the accuracy of its touchscreen dietary questionnaire.

As we noted in our paper (page 14, lines 5-6), the UK Biobank cohort is based on a sample of ‘healthy volunteers’, and comparisons have shown that people in UK Biobank are less likely to be obese, drink alcohol, or be current smokers than the UK general population. For example, compared to one national survey (the Health Survey for England), men and women in UK Biobank aged 45-54 were 7% and 9% less likely to be smokers, respectively [1].

Similar to this, as part of a larger study [2] we also compared dietary responses in UK Biobank to the National Diet and Nutrition Survey (NDNS), a nationally representative UK dietary survey, and found that those in UK Biobank were 8% more likely to report meeting the ‘5-a-day’ FV guideline than in the NDNS, at 38% and 30%, respectively.

Based on these patterns, one could estimate that values from the dietary questionnaire may be over- or under-reported by 5-10%, however it is impossible to know for certain in the absence of objectively measured data. Notably, another study that examined the performance of the UK Biobank dietary questionnaire has reported that participant responses for FV and meat consumption are very consistent over time (70-90%) and also correlate well with other independent dietary assessments (e.g. 24 hour dietary recall) conducted as part of the wider UKB cohort study [3]. This information about performance of the questionnaire has been added to the revised manuscript on page 14, lines 13 to 17:

“Due to the health-conscious nature of the cohort, it is possible that consumption of some food groups may be over- or under-reported, however, an in-depth study of the reliability of the UKB touchscreen dietary questionnaire has shown that participant responses for FV and meat consumption are very consistent over time (70-90%) and correlate well with other independent dietary assessments (e.g. 24 hour dietary recall) conducted as part of the larger UKB study.”

Comment #6: Hypothesis: in a heavily car-dominated transport environment, those who cycle are likely to deviate from social norms in other respects (eg smoking behavior, dietary choices). But self-selection of this kind is less marked when the prevalence of cycling is high. Was it possible to stratify by region, comparing the associations between active transport and diet in high-cycling and low-cycling areas of the UK?

Author response: We agree with the reviewer’s hypothesis that cyclists are less likely to deviate from social norms in places where rates of cycling are truly high, such as the Netherlands (e.g. 27% of all trips are cycled) [4]. Based on current evidence, however, there are few, if any, places that achieve a comparably high rate of cycling in the UK [5], so most UK cyclists are still travelling in a car-dominated environment and are thus likely to be different from others in the population. Though there are some areas in the UK with relatively high rates of cycling (e.g. Cambridge, where 54% cycle at least once per week) [5], it was not possible to stratify our results to these specific areas based on the data we had available (e.g. no UK Biobank assessment centres were located in the Cambridge area).

However, we have added 2 sentences to the revised manuscript that suggests this as a direction for future research on page 15, starting at line 23:

“Since cycling is still a relatively rare form of travel in the UK, these patterns may reflect the fact that people who cycle for transport are somewhat unique, and may also deviate from social norms in other ways (e.g. diet). Future research could explore this area further by examining whether relationships between cycling and dietary consumption are consistent in parts of the UK where people cycle at higher frequencies (e.g. Cambridge) [61] or among those who cycle at higher intensities, such as for sport.”

Comment #7: Was it possible to distinguish between sports cyclists and others who ride bicycles, at less intense levels? Particularly for non-work journeys, this may be relevant.

Author response: We agree that this could be a useful distinction, however, it was not possible to distinguish sport cyclists from transport cyclists in the UK Biobank cohort. There was also no information collected on distance travelled or time spent in non-work cycling journeys. As shown in the comment above, we have added a sentence to the revised manuscript that suggests this as a direction for future research on page 16, line 2.

REVIEWER 2

Comment #1: This is a very interesting and well written manuscript, exploring the associations between active travel and a healthy, low carbon diet. Novel data is presented indicating an association between active travel participation, especially cycling, and increased consumption of fruit and vegetables and reduced consumption of red and processed meat.

Author response: Thank you for reviewing our manuscript and for your positive comments on the paper as a whole.

Comment #2: The abstract was heavily weighted to the results section but I think this is appropriate for such an article that is drawing on already existing data that has had the design, setting and participants described in detail and published elsewhere.

Author response: We agree that the abstract is heavily weighted to the results section but feel that it is appropriate in this case for the reasons the reviewer has already stated. Nevertheless, we have added a few additional details to the abstract on the exposure measures that we felt were relevant to the study, page 2 lines 16-17:

“Exposure measures Mutually exclusive mode or mode combinations of travel (car, public transport, walking, cycling) for non-work and commuting journeys.”

REVIEWER 3

Comment #1: Dear authors,

As a statistical reviewer of your manuscript I would like to congratulate you on a job well done. You have performed an analysis of a rather complex dataset with categorical measures as both outcomes and exposures. This calls for a rather advanced analytical approach and, in my judgement, you have successfully applied the appropriate methods using an ordinal logistic regression model. There are some decisions to be made in such a situation, e.g. the proportional odds assumption, and you have described these and provided arguments and reasoning for each decision and addressed potential violations. The results from the analyses are presented in great detail in tables and summarized in a sensible way in the text and conclusions. You also show due care not to infer any causality from the results. As such, I have no real reservations about the statistical methods presented in your manuscript.

The chosen statistical model might be unfamiliar to some readers and they might benefit from an added sentence that describes exactly how to interpret the reported odds ratios from a proportional odds ordered logistic regression model.

Author response: We thank this reviewer for his positive comments on our statistical analysis and helpful feedback. We have taken his advice and added 2 sentences to the statistical analysis section (pg 9, starting at line 24) that better explain how to interpret the odds ratios in a proportional odds model:

“When interpreting the ordinal logistic model, the model assumes that the relationship between each pair of outcome groups is the same, or in other words, that the coefficients describing the relationship between the lowest outcome category and all higher categories are the same as those describing the relationship between the next lowest category and all higher categories, etc. This is called the proportional odds or parallel lines assumption [45], and in this case, the models assume that the odds of being in the lowest dietary consumption category compared to the two highest, are the same as the odds of being in the highest consumption category compared to the two lowest.”

References

1. Fry A, Littlejohns TJ, Sudlow C, et al. Comparison of Sociodemographic and Health-Related Characteristics of UK Biobank Participants with the General Population. *Am J Epidemiol* 2017 doi: 10.1093/aje/kwx246[published Online First: Epub Date]].
2. Smith M. Prevalence, patterning, and predictors of health-and climate-relevant lifestyles in the UK: A cross-sectional study of travel and dietary behaviour in two national datasets. University of York, 2018.
3. Bradbury KE, Young HJ, Guo W, et al. Dietary assessment in UK Biobank: an evaluation of the performance of the touchscreen dietary questionnaire. *Journal of nutritional science* 2018;7:e6 doi: 10.1017/jns.2017.66[published Online First: Epub Date]].
4. Harms L, Kansen M. *Cycling Facts*. The Hague: Netherlands Institute for Transport Policy Analysis (KIM), 2018.
5. DfT. *Walking and Cycling Statistics, England: 2017*. London Department for Transport, 2018.

VERSION 2 – REVIEW

REVIEWER	Alistair Woodward University of Auckland New Zealand
REVIEW RETURNED	28-Jul-2019
GENERAL COMMENTS	Thank you for the opportunity to read the revised version of the paper. I believe the questions I raised initially have been answered satisfactorily, the paper is considerably improved this time round, and I have no further concerns.